# OpenReview forum: "AgentXploit: End-to-End Red-Teaming for AI Agents Powdered by Multi-Agent Systems"
_ICLR.cc/2026/Conference — ICLR 2026 Conference Withdrawn Submission_

### Official Review · Reviewer_mYgZ · 2025-10-29

**Soundness:** 3
**Presentation:** 2
**Contribution:** 2
**Rating:** 2
**Confidence:** 3

**Summary:**

The authors construct an automated red-teaming setup for generic agents, focusing on coding, research, and scheduling agents in their tests.  They assume complete white-box access to the agent's API and documentation throughout their attacks.  Red-teaming occurs in two steps: first, an analyzer identifies potential vulnerabilities and then an exploiter develops attacks to exploit these capabilities.  AgentXploit achieves higher ASRs than past agentic red-teaming efforts.

**Strengths:**

Agents comprise a higher and higher fraction of LLM deployments, so this is clearly a very important topic.  AgentXploit produces red-teaming methods that are automated, scalable, and general.  This could save significant red-teaming effort and ultimately lead to safer agents.

The structure of using an analyzer and then an exploiter is somewhat innovative.

The paper is clearly written and the engineering effort seems well-executed.

**Weaknesses:**

1.  There are issues with motivation in this paper.  Of course, red-teaming is important, but if the goal is to produce a generic red-teaming framework, then several things don't make sense.  How can one generically measure attack success?  There are predefined notions of ASR in AgentDojo, but if I were to apply this to a new agent, how do we easily measure ASR?  Additionally, the white-box setting doesn't entirely make sense to me.  Having access to documentation about the API calls seems reasonable, but injection into accessed documents really doesn't make sense.  Sure, you can upload web docs that have embedded attacks, but how can you guarantee that the agent will come across those?  It seems like the paper is too liberal in the access that it hands to the attacker.
2.  Scientifically, I don't understand the contribution of this paper.  Sure, you hill climb on ASRs, but I don't see any scientific contribution or novelty.  Prompt injection attacks, agentic red-teaming, and automated red-teaming are all well-studied.  What makes this paper a scientific feat rather than just a (very well-executed) engineering effort?
3.  The stealthiness of this red-teaming also seems like an important and under-explored factor.  Papers like [1, 2] study fine-tuning attacks in agentic and non-agentic settings, but they measure not only ASR, but also the detectability of the attacks.  It seems important to check that your attacks don't lower the utility of the agent too much (as in [1]), otherwise they are very easily detected and counter-measures can be quickly taken.

[1] https://openreview.net/forum?id=RwoMf7YSfD
[2] https://arxiv.org/abs/2502.19537

**Questions:**

I don't understand your definition of white-box access in this paper.  Typically this means that you have access to the model weights and API calls.  In line 316, you state "the primary attack path lies in malicious content injection through reference documents".  How are you able to alter the documents that the API calls retrieve?  This seems like it goes beyond white-box access.  Won't the model select documents from the web that you don't necessarily have the capability to edit?

Similarly, I don't understand what threat model would allow for an attack like the one described starting on line 343

---

### Official Review · Reviewer_1QBs · 2025-11-01

**Soundness:** 3
**Presentation:** 2
**Contribution:** 3
**Rating:** 4
**Confidence:** 4

**Summary:**

This paper introduces AGENTXPLOIT, an end-to-end red-teaming framework that uses a multi-agent system (Attacker Agent vs. Victim Agent) to automatically identify and exploit complex attack paths within AI agents that leverage external tools and data sources. The core innovation lies in shifting the focus from generating individual attack payloads to systematically constructing the path from a system's entry point to a vulnerable component, addressing a major gap in current agent security research.

**Strengths:**

1. Addresses a high-priority, evolving security threat (IPI) in the most complex form of LLM deployment (multi-tool, multi-agent systems).

2. The use of a multi-agent red-teaming framework (AGENTXPLOIT) is a novel and interesting approach to solving the graph search problem of attack path discovery.

3.  The Exploiter agent achieves sota results on the established AgentDojo benchmark a ~79% Attack Success Rate (ASR).

4. The ablation studies clearly isolate and quantify the distinct contributions of the Exploiter's core components, which is critical for the research community.

**Weaknesses:**

1. Lacks sufficient analysis of the framework's limitations and failure modes.

2. While the paper compares the Exploiter against previous SOTA prompt injection methods, it fails to provide a strong baseline for the core challenge: automated path discovery by the Analyzer. There is no comparison against non-LLM-based algorithms.

3. The explicit assumption of complete white-box access to the target agent's codebase and documentation is a severe limitation on the framework's direct utility. Most critical commercial LLM agents are closed-source.

**Questions:**

N/A

---

### Official Review · Reviewer_g39s · 2025-11-03

**Soundness:** 3
**Presentation:** 2
**Contribution:** 2
**Rating:** 4
**Confidence:** 4

**Summary:**

This paper introduces AgentXploit, a fully automated, multi-agent red-teaming framework for systematically discovering and exploiting vulnerabilities in LLM-powered AI agents. The framework operates in two stages: an Analyzer agent conducts white-box static and dynamic analysis on the agent’s codebase to uncover high-risk tools and feasible attack paths, and an Exploiter agent leverages these findings to craft and execute context-aware, iterative indirect prompt injection attacks. The methodology is evaluated on popular open-source agents (OpenHands, GPT-Researcher) and the AgentDojo benchmark, achieving a 79% attack success rate, significantly outperforming prior state-of-the-art. The paper includes detailed ablation studies and qualitative analyses of attack examples.

**Strengths:**

- **Systematic, End-to-End Red-Teaming**: The clear two-stage design (Analyzer and Exploiter) represents a valuable step toward automating real-world vulnerability discovery and exploitation in LLM agents, bridging a gap in prior work which assumed given attack paths.
- **Substantive Empirical Results**: On AgentDojo, AgentXploit achieves a 79% attack success rate, representing a 27% improvement over the state-of-the-art AgentVigil. The system is further validated through end-to-end case studies across practical agent types.
- **Relevance and Timeliness**: The topic is timely, targeting the increasing risk profile of tool-augmented LLM agents in real applications and providing actionable insights for the community.

**Weaknesses:**

- **Limited Scope for Black-Box Scenarios**: The approach presumes white-box access to source code and documentation, which, while valuable for developer-side red-teaming, limits external applicability. More discussion is needed on partial information or black-box adversary settings, especially since many high-impact attacks may target proprietary, closed-source agents. This is noted but not deeply explored in Section 3 or the Discussion.
- **Potential Bias in Experimental Evaluation and Generalization**:
The evaluation on just two major open-source agents (OpenHands, GPT-Researcher) and the AgentDojo framework, while relevant, may not fully represent the diversity of agent architectures encountered in the wild. No results are reported for commercial or multi-modal agents, and there’s little discussion about scenario transfer or failure cases. Also, how does the system handle dynamically loaded tools or non-standard, non-Python based agents?
Table 1 and ablations show strong improvements on the chosen benchmarks, yet it remains unclear how well these translate to “defended” agents that implement up-to-date mitigations.
- **Reproducibility/Transparency of Analyzer Implementation**: The Analyzer’s integration with LSP/MCP, details of context compaction mechanisms, and specific heuristics for code/task prioritization are described at a high level in Section 4.1, but full pseudo-code or detailed algorithmic breakdowns are missing. This hinders precise reproduction of results.
- **Missing Recent Directly Related Work**: Several highly relevant recent efforts (on black-box red-teaming [1], multimodal attacks [2], GUI agent attacks [3], and multi-agent protocol exploitation [4]) are not discussed. This weakens claims regarding novelty, especially as the field progresses rapidly.

---
References:

[1]. Wang, Z., Siu, V., Ye, Z. (2025): "AgentXploit: End-to-End Redteaming of Black-Box AI Agents"
[2]. Li, Y., Cao, Y., Wang, D. (2025): "AgentTypo: Adaptive Typographic Prompt Injection Attacks against Black-box Multimodal Agents"
[3]. Lu, Y., Ju, T., Zhao, M. (2025): "EVA: Red-Teaming GUI Agents via Evolving Indirect Prompt Injection"
[4]. He, P., Lin, Y., Dong, S. (2025): "Red-Teaming LLM Multi-Agent Systems via Communication Attacks"

**Questions:**

Success Criteria/Definition: Could the authors formalize exact criteria for exploit “success” in both synthetic (AgentDojo) and real-world open-source agents? Is there a mathematically precise description (e.g., unauthorized output/action quantification) beyond qualitative/validator-based judgments?

Limits of Current Implementation: What are the key bottlenecks or limitations (e.g., context length, LLM hallucination, coverage tradeoffs) that the authors encountered in their end-to-end automation? Are there scenarios where Analyzer/Exploiter fails silently?

Reproducibility and Implementation Details: Can the authors provide concrete pseudo-code, pipelines, or more formal specifications for the Analyzer’s context management? Increased algorithmic specificity would help with reproducibility.

Generalization Beyond Python/Open-Source Agents: Can the authors clarify or empirically characterize the framework’s robustness to diverse or obfuscated agent implementations?

---

### Note · Authors · 2025-11-24

I have read and agree with the venue's withdrawal policy on behalf of myself and my co-authors.